# Cancer-Related Pain Management in Suitable Intrathecal Therapy Candidates: A Spanish Multidisciplinary Expert Consensus

**Concha Pérez** [1,*] , **Teresa Quintanar** [2] , **Carmen García** [3], **Miguel Ángel Cuervo** [4] , **María Jesús Goberna** [5], **Manuela Monleón** [6], **Ana I. González** [7], **Luís Lizán** [8] , **Marta Comellas** [8] , **María Álvarez** [9] **and Isaac Peña** [10]

1   Hospital Universitario de la Princesa, 28006 Madrid, Spain
2   Hospital General Universitario de Elche, 03203 Elche, Spain; teresaqv22@yahoo.es
3   Unidad de Continuidad Asistencial, Servicio Madrileño de Salud, 28046 Madrid, Spain; cgcubero@salud.madrid.org
4   Hospital Universitario de Badajoz, 06080 Badejoz, Spain; miguelangel.cuervop@gmail.com
5   Complejo Hospitalario Universitario de Vigo, 36204 Vigo, Spain; maria.jesus.goberna.iglesias@sergas.es
6   Equipo de Soporte de Atención Domiciliaria de Legazpi, 28045 Madrid, Spain; manuela.monleon@salud.madrid.org
7   Asociación Española Contra el Cáncer (AECC), 28045 Madrid, Spain; ana.gonzalez@contraelcancer.es
8   Outcomes'10, Departamento de Medicina, Universidad Jaume I, 12071 Castellón, Spain; lizan@outcomes10.com (L.L.); mcomellas@outcomes10.com (M.C.)
9   Health Economics & Outcomes Research Unit, Medtronic Ibérica, S.A., 28050 Madrid, Spain; maria.alvarez.orozco@medtronic.com
10  Hospital Universitario Virgen del Rocio, 41013 Sevilla, Spain; isaacpv@gmail.com
*   Correspondence: concha.phte@gmail.com; Tel.: +34-915-20-22-00

**Abstract:** A consensus is needed among healthcare professionals involved in easing oncological pain in patients who are suitable candidates for intrathecal therapy. A Delphi consultation was conducted, guided by a multidisciplinary scientific committee. The 18-item study questionnaire was designed based on a literature review together with a discussion group. The first-round questionnaire assessed experts' opinion of the current general practice, as well as their recommendation and treatment feasibility in the near future (2–3-year period) using a 9-point Likert scale. Items for which consensus was not achieved were included in a second round. Consensus was defined as ≥75% agreement (1–3 or 7–9). A total of 67 panelists (response rate: 63.2%) and 62 (92.5%) answered the first and second Delphi rounds, respectively. The participants were healthcare professionals from multiple medical disciplines who had an average of 17.6 (7.8) years of professional experience. A consensus was achieved on the recommendations (100%). The actions considered feasible to implement in the short term included effective multidisciplinary coordination, improvement in communication among the parties, and an assessment of patient satisfaction. Efforts should focus on overcoming the barriers identified, eventually leading to the provision of more comprehensive care and consideration of the patient's perspective.

**Keywords:** intrathecal; cancer; pain; Delphi technique; consensus

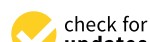



## 1. Introduction

Pain is one of the most common and distressing symptoms for cancer patients [1], and its treatment is a significant factor affecting quality of life [2]. Pain prevalence rates range from 39% in patients following curative treatment to 66–80% in patients in advanced phases [3]. Epidemiological studies carried out in Spain have shown that approximately 55% of cancer patients suffer from pain [4].

According to the WHO's analgesic ladder, the strategy for cancer pain relief begins with conservative options, such as pharmacological treatment based on opioids or other

pharmaceuticals and nonpharmacological interventions, prior to more aggressive and invasive interventions [5]. However, it is estimated that from 10 to 20% of patients are unresponsive or unable to tolerate the conventional first-line approaches and, therefore, require interventional treatment [6]. Interventional analgesic therapies can be divided into five basic classes: neuraxial analgesia, minimally invasive procedures for vertebral pain, sympathetic axis blocks for abdominal cancer pain, peripheral nerve blocks, and percutaneous cordotomy [7]. In this study, we focused exclusively on those patients who benefit from neuraxial therapy by intrathecal route, since the epidural route is reserved for patients with a shorter life expectancy (<3 months) [7].

The advantages of intrathecal therapy include the continual administration of analgesics directly at their site of action [8]; increased efficacy of pain management with an associated reduction in side effects compared with systemic administration [9]; and improved measures of fatigue and alertness, resulting in improved quality of life (for instance, reducing constipation associated with other treatments [10]). Furthermore, intrathecal therapy reduces costs related to health resource consumption derived from other treatments (such as reduced use of outpatient resources and decreased prescription of medications) [8]. However, intrathecal therapy is not without risk. These complications can be procedure related, device specific, or associated with an administered drug and can range from catheter-tip granuloma to infections such as meningitis [11].

Suitable patient selection is essential for the optimal management of intrathecal therapy [9,10]. It should take into consideration a patient's treatment history with special attention to concomitant therapies and their adverse events. Furthermore, comorbid psychiatric disorders and psychosocial issues that could negatively affect the treatment outcome should be adequately addressed [12,13]. It is essential to guarantee patient's psychological well-being, social support structure, healthcare coverage, finances, and probability/capability of adherence to intrathecal therapy requirements [9,13]. On the other hand, the decision to implement this type of analgesia in a patient must consider the risk–benefit ratio, choosing the most appropriate moment to improve pain management and the patient's quality of life compared to the risks entailed by the necessary surgery [12]. Intrathecal therapy has been successfully applied in home care patients, where the patients' and caregivers' knowledge of the system and its proper usage are guaranteed, along with the experience and qualification of the healthcare professionals involved in their follow-up [14]. Altogether, the multidisciplinary coordination and communication processes among oncology teams, pain units and, in due course, palliative care units are essential.

Traditionally, intrathecal therapy was only considered for patients with refractory or complex cancer-related pain, including individuals at the end of life [13]. In the last decade, however, it has been recognized as a beneficial therapy for patients who do not obtain adequate analgesia after a reasonable course of systemic opioid treatment and in those with dose-limiting side effects [10,12]. Thus, intrathecal therapy can be the therapeutic option of choice for cancer patients at any time during the course of the disease, always prioritizing its adequate timing to avoid unnecessary suffering, especially for those patients with limited life expectancy [10,13].

Despite the aforementioned advantages and the numerous algorithms in place to treat oncological pain including intrathecal therapy in the services portfolio [15], several limitations still prevent its uniform implementation at all levels [16,17]. Some of the identified barriers include the insufficient number of qualified healthcare professionals with adequate training in cancer and related treatments, including both pharmacological and invasive anesthetic techniques [18], the optimal healthcare circuit in place for the medical teams involved [19], or a misperception of the risk–benefit ratio between surgery and analgesic duration [16]. Moreover, delayed referrals from primary care to specialized pain units could pose a limitation to implementing interventional techniques due to the advanced stage of the disease [18].

Our study aimed to establish a consensus among healthcare professionals in the different units involved in oncological pain management in patients who are suitable candidates to receive and benefit from intrathecal therapy. The survey addresses the optimal protocol of action and healthcare circuit that would guarantee the best strategy for cancer pain relief, the ideal patient selection criteria for this therapy, the preferred clinical and patient-reported follow-up variables, and the healthcare quality indicators of choice to achieve continuous quality control of the care offered.

## 2. Materials and Methods

The study was led by a multidisciplinary scientific committee of experts in cancer patient management. It consisted of one physician and one nurse from the pain unit, one physician and one nurse from the palliative care unit, one oncologist, one case manager, and one patient representative from the Spanish Association Against Cancer ($n = 8$). The study comprised three phases: (1) literature review, (2) discussion group, and (3) a Delphi consultation.

### 2.1. Literature Review

A literature review was conducted in the international databases PubMed/Medline and national databases IBECS/MEDES to identify existing evidence on the management of cancer patients receiving intrathecal therapy regarding the protocol of action or healthcare circuit, the patient selection criteria, the variables and instruments used for their monitoring, and the healthcare quality indicators.

### 2.2. Discussion Group

A discussion group was held with the scientific committee in order to revise the information provided by the literature review, define the aspects of oncological pain management to be explored in the Delphi consultation, and design the Delphi questionnaire.

### 2.3. Delphi Consultation

The Delphi method is a formal and systematic approach to obtain consensus from a group of experts by means of a series of short, self-administered questionnaires [20].

A two-round Delphi was prepared. The first-round questionnaire consisted of sociodemographic variables and 18 statements grouped in four sections: protocol of action and healthcare circuit ($n = 8$ statements), patient selection criteria ($n = 2$ statements), clinical and patient-reported follow-up variables ($n = 3$), and healthcare quality indicators ($n = 5$ statements). In each statement posed, the experts were asked to express their opinion on the current situation of the standard general practice, as well as their recommendation and the feasibility of implementation in the near future (i.e., 2–3-year period). Each question was scored on a 9-point Likert scale (1, strongly disagree; 2, disagree; 3, moderately disagree; 4, slightly disagree; 5, neither agree nor disagree; 6, slightly agree; 7, moderately agree; 8, agree; 9, strongly agree), corresponding to the extent to which the expert agreed with the item being addressed. Participants were also provided with a free-text space at the end, in which they could make observations and comments. The second-round questionnaire comprised the statements for which no consensus was reached in the first round, concerning their recommendation and feasibility in the near future (2–3-year period). Their perspective regarding the current situation was not considered for consensus, as it aimed to describe the present standard general practice. The second-round questionnaire consisted of eight statements and was specifically tailored to each expert. Every statement contained information regarding the score he/she recorded in the first round and the position of the overall group (range of the greatest percentage of scores). Each expert was invited to confirm his/her position or modify the score in order to bring it closer to that of the group so that a consensus could be reached on the greatest possible number of statements.

Healthcare professionals with extensive experience in cancer pain management were selected as panelists and invited to participate. Pain unit anesthesiologists, oncologists,

palliative care specialists, nurses involved in cancer pain treatment (including hospital and community nurses), home hospitalization unit specialists (including primary care physician), and case managers were included.

The invitation to the first-round electronic questionnaire was sent by email and remained active between May and June 2022 and the second round between June and July 2022.

*2.4. Data Analysis*

In order to identify the statements that reached a consensus in both rounds, the 9 response options were grouped into three categories: rejection—1, strongly disagree; 2, disagree; 3, moderately disagree; indetermination—4, slightly disagree; 5, neither agree nor disagree; 6, slightly agree; and agreement—7, moderately agree; 8, agree; 9, strongly agree.

Consensus was achieved for the recommendation and feasibility perspectives for each of the proposed question when at least 75% of participants scored in the range of rejection (1, 2, and 3) or agreement (7, 8, and 9). When no consensus was reached, the proposed question remained undetermined, as the participants' stance was unclear.

## 3. Results

*3.1. Panelists*

A total of 106 experts were invited to participate in the Delphi consultation. Sixty-seven fully answered the first-round questionnaire (response rate: 63.2%), and sixty-two answered the second-round questionnaire (response rate: 92.5%).

The panelists' sociodemographic characteristics are described in Table 1. The participants were 55.2% female, between 35 and 66 years old, and had between 5 and 40 years of professional experience in cancer pain management. Their medical discipline was anesthesiology (31.3%), nursing (23.9%), oncology (19.4%), palliative care (19.4%), and case management (6.0%). The study targeted professionals practicing in all Spanish regions, with greatest representation corresponding to Madrid (19.4%), Galicia (19.4%), and Andalucía (14.9%), while no results were obtained from La Rioja, Navarra, Ceuta and Melilla, Aragón, Asturias, or Baleares (see detail in Table S1 in the Supplementary Materials).

**Table 1.** Sociodemographic characteristics of participants (*n* = 67).

| Characteristics | Value |
|---|---|
| **Sex**, female, % (*n*) | 55.2 (37) |
| **Age**, years, (mean (SD)) | 51.8 (8.0) |
| **Time of professional experience**, years, (mean (SD)) | 17.6 (7.8) |
| **Medical discipline**, % (*n*) | |
| Anesthesiology | 31.3 (21) |
| Nursing | 23.9 (16) |
| Oncology | 19.4 (13) |
| Palliative care | 19.4 (13) |
| Case managers | 6.0 (4) |

*3.2. Current Situation and Consensus*

Consensus was achieved in the 18 proposed statements (100%) from the recommendation perspective, and 12 (66.7%) from the feasibility perspective. Table 2 shows the full set of results, including perspectives on current standard clinical practice.

3.2.1. Protocol of Action and Healthcare Circuit

In line with the current standard clinical practice, the experts reached a consensus on both the recommendation and feasibility perspectives for patients who are intrathecal therapy candidates to be referred to the pain unit, where decisions are taken on the appropriate combination of pharmaceuticals to be administered via intrathecal route and the dose. In addition, experts also agreed that nursing professionals from the pain unit should instruct

patients and their families or caregivers on the proper usage and care of the system after the implantation procedure and before hospital discharge.

**Table 2.** Delphi consultation results (*n* = 67).

| Statement | % | | |
|---|---|---|---|
| | **R** | **I** | **A** |
| **PROTOCOL OF ACTION AND HEALTHCARE CIRCUIT** | | | |
| 1. **In the management of patients with cancer pain who are suitable candidates for intrathecal therapy, a multidisciplinary/comprehensive approach is applied, including clinical and psychosocial specialties.** | | | |
| • Present | 25.4 | 31.3 | 43.3 |
| • Recommendation | 4.5 | 6.0 | **89.6** |
| • Feasibility | 11.3 | 21.0 | 67.7 |
| 2. **Patients with cancer pain who are suitable candidates for intrathecal therapy receive formal psychological evaluation and psychological support is offered only when the patient requires it throughout the care process.** | | | |
| • Present | 38.8 | 34.3 | 26.9 |
| • Recommendation | 4.5 | 9.0 | **86.6** |
| • Feasibility | 11.3 | 32.3 | 56.5 |
| 3. **Patients with cancer pain who are suitable candidates for intrathecal therapy are referred by their healthcare professional (or by a decision-making committee, if applicable) to the Pain Unit.** | | | |
| • Present | 9.0 | 16.4 | 74.6 |
| • Recommendation | 3.0 | 1.5 | **95.5** |
| • Feasibility | 6.0 | 3.0 | **91.0** |
| 4. **The Pain Unit oversees the decision concerning the appropriate combination of pharmaceuticals and the dose to be administered via intrathecal route.** | | | |
| • Present | 4.5 | 3.0 | **92.5** |
| • Recommendation | 3.0 | 0 | **97.0** |
| • Feasibility | 4.5 | 0 | **95.5** |
| 5. **Pain Unit nurses instruct patients and their families or caregivers on the required care of the intrathecal therapy system and its proper usage after the implantation procedure and before their discharge from the hospital.** | | | |
| • Present | 10.4 | 14.9 | 74.6 |
| • Recommendation | 1.5 | 0 | **98.5** |
| • Feasibility | 6.0 | 3.0 | **91.0** |
| 6. **There is a sufficient number of qualified healthcare professionals who are experts in pain management and who can provide interventional treatments.** | | | |
| • Present | 41.8 | 28.4 | 29.9 |
| • Recommendation | 3.0 | 13.4 | **83.6** |
| • Feasibility | 11.3 | 33.9 | 54.8 |
| 7. **The referral process/oncological pain patient support is quick (<48 h for urgent patients or <1 week for preferent patients), which allows for the optimization and guarantee of speedy access to care.** | | | |
| • Present | 31.3 | 23.9 | 44.8 |
| • Recommendation | 3.0 | 4.5 | **92.5** |
| • Feasibility | 11.3 | 16.1 | 72.6 |

**Table 2.** *Cont.*

| Statement | % | | |
|---|---|---|---|
| | **R** | **I** | **A** |
| 8. **An effective coordination among multidisciplinary healthcare professionals involved in the management of patients receiving intrathecal therapy is in place.** | | | |
| • Present | 26.9 | 31.3 | 41.8 |
| • Recommendation | 1.5 | 3.0 | **95.5** |
| • Feasibility | 6.5 | 17.7 | **75.8** * |
| **PATIENT SELECTION CRITERIA** | | | |
| 9. **Patients with refractory cancer pain or who are intolerant to noninvasive conventional treatment are suitable candidates to receive intrathecal therapy, despite not qualifying for palliative care.** | | | |
| • Present | 9. | 9.0 | **82.1** |
| • Recommendation | 1.5 | 3.0 | **95.5** |
| • Feasibility | 3.0 | 16.4 | **80.6** |
| 10. **A comprehensive assessment of the patient, including the evaluation of his/her functional state, life expectancy, comorbidities, or other psychosocial factors, must be considered in order to determine his/her suitability to receive intrathecal therapy.** | | | |
| • Present | 4.5 | 7.5 | **88.1** |
| • Recommendation | 1.5 | 3.0 | **95.5** |
| • Feasibility | 3.0 | 7.5 | **89.6** |
| **CLINICAL AND PATIENT-REPORTED FOLLOW-UP VARIABLES** | | | |
| 11. **During the follow-up of patients receiving intrathecal therapy, the treatment efficacy (including pain intensity, location or frequency, functional scale or quality of life, etc.) and the implant safety (adverse events) are evaluated and recorded in their clinical history.** | | | |
| • Present | 6.0 | 20.9 | 73.1 |
| • Recommendation | 1.5 | 4.5 | **94.0** |
| • Feasibility | 1.5 | 16.4 | **82.1** |
| 12. **The follow-up of patients receiving intrathecal therapy is personalized with the frequency or modality adapted to their needs.** | | | |
| • Present | 7.5 | 14.9 | **77.6** |
| • Recommendation | 1.5 | 3.0 | **95.5** |
| • Feasibility | 4.5 | 9.0 | **86.6** |
| 13. **During the follow-up of patients receiving intrathecal therapy, his/her access to the healthcare circuit is facilitated, for instance, with a direct-contact telephone number or an open-door consultation with the medical team.** | | | |
| • Present | 4.5 | 14.9 | **80.6** |
| • Recommendation | 1.5 | 1.5 | **97.0** |
| • Feasibility | 4.5 | 10.4 | **85.1** |

**Table 2.** *Cont.*

| Statement | % | | |
|---|---|---|---|
| | **R** | **I** | **A** |
| **HEALTHCARE QUALITY INDICATORS** | | | |
| 14. **Patients with cancer pain who are suitable candidates or who receive intrathecal therapy and/or their families or caregivers are informed verbally or receive written brochures with further information, additional to that included in the informed consent sheet about their treatment plan and the possible adverse events.** | | | |
| • Present | 11.9 | 34.3 | 53.7 |
| • Recommendation | 1.5 | 6.0 | **92.5** |
| • Feasibility | 4.5 | 17.9 | **77.6** |
| 15. **The treatment plan for patients with cancer pain who receive intrathecal therapy is agreed on with the patient and/or family.** | | | |
| • Present | 7.5 | 26.9 | 65.7 |
| • Recommendation | 1.5 | 3.0 | **95.5** |
| • Feasibility | 4.5 | 14.9 | **80.6** |
| 16. **The healthcare professionals involved in the management of patients with cancer pain receiving intrathecal therapy are trained in communication skills to improve their communication with the patients and their families or caregivers.** | | | |
| • Present | 22.4 | 41.8 | 35.8 |
| • Recommendation | 1.5 | 4.5 | **94.0** |
| • Feasibility | 3.2 | 27.4 | 71.6 |
| 17. **The satisfaction of the patient receiving intrathecal therapy and/or their family/caregivers with the care received is evaluated.** | | | |
| • Present | 13.4 | 46.3 | 40.3 |
| • Recommendation | 0 | 7.5 | **92.5** |
| • Feasibility | 6.5 | 14.5 | **79.0 *** |
| 18. **There are care and support protocols in place for patients with cancer pain who are suitable candidates to or who receive intrathecal therapy.** | | | |
| • Present | 23.9 | 43.3 | 32.8 |
| • Recommendation | 1.5 | 9.0 | **89.6** |
| • Feasibility | 9.7 | 21.0 | 69.4 |

R: rejection (1, strongly disagree; 2, disagree; 3, moderately disagree); I: undetermined (4, slightly disagree; 5, neither agree nor disagree; 6, slightly agree); A: agreement (7, moderately agree; 8, agree; 9, strongly agree); * consensus reached in the 2nd round.

Furthermore, experts agreed on the recommendation of effective coordination among healthcare professionals involved in the management of patients receiving intrathecal therapy and considered it feasible, even though it is not currently being conducted.

Finally, a consensus was reached on the recommendation of a multidisciplinary approach to patients with cancer-related pain, their timely referral and care (<48 h for urgent and <1 week for preferential patients), the possibility of providing a psychological assessment or emotional support if required, and having enough qualified healthcare professionals to assist them. Unfortunately, these actions are not being implemented currently, and a consensus was not reached on whether they are feasible to implement in the near future (2–3 years).

### 3.2.2. Patient Selection Criteria

Experts positively agreed on recommending patients with refractory cancer pain or who are intolerant to noninvasive conventional treatment as suitable candidates to receive

intrathecal therapy, despite not qualifying for palliative care. They also considered the above to be feasible. The same result was obtained for patient selection criteria, with agreement that a comprehensive assessment of the patient, including the evaluation of his/her functional state, life expectancy, comorbidities, or other psychosocial factors, should be considered in order to determine his/her suitability to receive intrathecal therapy. Fortunately for oncological patients with cancer-related pain, these two actions are already part of the current standard clinical practice.

### 3.2.3. Clinical and Patient-Reported Follow-Up Variables

The experts reached a full consensus, agreeing on the recommendation and feasibility of performing a personalized follow-up of patients receiving intrathecal therapy. This should include evaluating and recording the efficacy of the treatment (namely, pain intensity, location or frequency, functional scale or quality of life, etc.) and implant safety (adverse events) in their clinical history, as well as facilitating patients' access to the healthcare circuit during their follow-up (for instance with a direct-contact telephone number or open-door consultation with the medical team). The current standard clinical practice includes personalized follow-up and easy access to the healthcare circuit; however, the experts considered that clinical history recording could be improved.

### 3.2.4. Healthcare Quality Indicators

The actions in this section are not generally performed in current standard clinical practice. Nonetheless, the experts reached a consensus, agreeing on the recommendation and feasibility that suitable candidates for intrathecal therapy and/or their families should be informed verbally or receive information brochures with additional information to that provided in the informed consent sheet on their treatment plan and possible adverse events. Consensus was reached that the treatment plan should be agreed on with the patient and/or family and that their satisfaction with the attention received should be assessed.

Finally, the experts also agreed on recommending that the healthcare professionals managing this group of patients should be formally trained in communication skills to improve communication with patients and their families and that protocols should be put in place for better patient care and support. Neither of these actions are currently being performed and a consensus on whether their implementation is feasible in the near future (2–3 years) was not reached.

## 4. Discussion

This Delphi study, involving a multidisciplinary group of experts, assessed perspectives on the management of patients with oncological pain, who are suitable candidates to receive and benefit from intrathecal therapy. The information gathered provides insights into improving pain management in these patients.

The recommendation for interventional procedures throughout cancer patients' care should be considered as an integrative approach, combined with other measures rather than an alternative following the failure of other analgesic options [21]. Several studies have shown how interventional procedures in cancer pain management provide positive outcomes not only by relieving the pain but also by reducing other symptoms that directly affect the patients' well-being and decreasing daily opioid use [8–10,13,21].

Experts unanimously agreed that the coordination and communication of multidisciplinary medical teams involved in cancer treatment and cancer pain management is essential to ensure that all relevant aspects of patient needs are considered. This holistic approach is even more important in the case of complex or refractory pain [22]. Thus, patients should be referred to pain specialists when pain does not improve quickly or intolerable side effects of analgesia are expected [22,23]. As previously reported [9], some progress has been made in recent years, but the implementation of this multidisciplinary approach is infrequent because of the complexity and dynamism of cancer pain [18], especially when considering both hospital and home care. A recent survey among medical

oncologists revealed that multidisciplinary pain management and collaboration with other specialists are still uncommon [21]. In fact, this lack of a multidisciplinary approach has been identified as a factor preventing optimal cancer pain management [17]. In addition, there is still a lack of real-world evidence regarding the interventional management of cancer pain, its risks, and benefits [23].

In this respect, establishing a case management system has helped, as it is essentially multidisciplinary and patient-centered, helping cancer patients navigate the healthcare system throughout their illness [24,25]. Alternatively, establishing healthcare centers that bring together multiple medical specialties, also known as pain clinics, can help improve coordination and treatment. However, because of the scarce availability of such centers, attempts are often made to optimize analgesic treatment based on the WHO's guidelines prior to referring patients to pain specialists [26]. Lastly, another proposal is the creation of a multidisciplinary oncological committee to assess pain treatment efficacy for each patient, making sure that the pain-relief strategy is the appropriate to their needs throughout their evolution, assuming the obligation to indicate and propose a change when necessary. From an organizational point of view, this coordination should include home-care programs, as it is preferable to treat patients' pain from the comfort of their home, whenever medically possible.

Regarding psychological assessment and support targeting patient needs, the experts agreed on recommending that professionals from this medical discipline should form part of the multidisciplinary team. They acknowledged that an initial assessment should be performed with the results being used as selection criteria. However, support is not always feasible during the follow-up; therefore, the development of an objective algorithm to help identify patients' psychological state is proposed.

Regarding the clinical and patient reported follow-up variables, the study highlights the need for systematized records to achieve efficacy and safety of the clinical history not only concerning the ongoing oncological treatment (namely, dose, adverse events, and medication being administered) but also other relevant records associated to comorbidities and the management of pain, as well as other symptoms.

The panelists were far less optimistic about the feasibility of implementing initiatives, because of, for example, the availability of a sufficient number of qualified healthcare professionals due the current employment system. One possible alternative proposed would be to incorporate nonexclusive personnel able to work for several services or areas when required. The need for life-long training of the healthcare professionals currently employed in pain units is also highlighted and should include not only medical training on cancer and its treatment, both pharmacological and invasive anesthetic, but also soft skills. Particularly, the need for improving communication between the healthcare professionals and patients and their families is recommended. Currently, medicine is becoming more patient centered, with experts recommending that patients should be better informed and more aware and, thus, able to participate actively in decisions regarding their treatment plan. In order to achieve this successfully, healthcare professionals should also show empathy towards their patients.

Lastly, regarding the healthcare quality indicators, experts agreed that relevant criteria should include an improvement in the patient's functionality, the number of visits to the pain unit, and the number of explants due to complications or a cost–benefit ratio analysis.

An important strength of the Delphi technique is the choice of an appropriate panel of participants. In this study, our aim was to obtain a sample representing all medical disciplines involved in oncological pain management. Moreover, the Delphi questionnaire was designed by a scientific committee and, thus, included the views of a multidisciplinary group of experts, who helped to define the appropriate and inappropriate approaches to addressing the gaps in current care.

This study presents several limitations inherent to the methodology; namely, the consensus was based on the participants' experience in the Spanish context and was set to at least a 75% level of agreement. This definition of consensus is that most com-

monly reported in the literature [27,28]; however, it is necessary to acknowledge that another definition could have led to different results. In addition, it is important to note that results should be extrapolated with caution to other people or places outside of the Spanish context.

## 5. Conclusions

The results of this study show a high degree of consensus among experts regarding the recommendations for the appropriate approach of patients with cancer-related pain, who are suitable candidates for intrathecal therapy. However, they considered that the implementation of some actions would not be feasible in the near future (2–3 years). These results can help to overcome the identified barriers and guide healthcare professionals in making decisions to achieve more comprehensive care in the context of administering intrathecal therapy and focus on improving the patient's experience, follow-up, and perspective.

**Supplementary Materials:** The following supporting information can be downloaded at: https://www.mdpi.com/article/10.3390/curroncol30080530/s1, Table S1: Number of professionals participating from the different Spanish regions.

**Author Contributions:** Conceptualization, C.P., T.Q., C.G., M.Á.C., M.J.G., M.M., A.I.G., L.L., M.C., M.Á. and I.P.; methodology, L.L. and M.C.; formal analysis, L.L. and M.C.; writing—original draft preparation, M.C.; writing—review and editing, C.P., T.Q., C.G., M.Á.C., M.J.G., M.M., A.I.G., L.L., M.Á. and I.P.; supervision, C.P. All authors have read and agreed to the published version of the manuscript.

**Funding:** This research was funded by Medtronic Iberica S.A.

**Institutional Review Board Statement:** Ethical review and approval were waived for this study because of its nature (Delphi consensus) and its participants (health professionals).

**Informed Consent Statement:** Not applicable.

**Data Availability Statement:** The data presented in this study are available upon request from the corresponding author.

**Acknowledgments:** We would like to sincerely thank all Delphi Panelists (C. Aguilera González, A. Alonso, L. Alonso Prieto, C. Álvarez, L. Ángel, M.A. Benítez-Rosario, Y. Camacho, L. Canovas, A. Carregal, G. Casado, J. Costillo, M.J. De la Fuente, A. De la Iglesia, R. De las Peñas, C. Dürsteller, FM. Estévez, A. Falcón, T. Fernández, I. Fernández, M. Fernández, J. Fernández, M. Ferreiro, J. Flores, D. Gainza, R. Gálvez, A. Gisbert, J. Gómez-Ulla Astray, J. Julià Torras, M. Labori Trias, D. Lasuen, N. López-Casero, A. Manzanas Gutiérrez, C. Margarit, M.A. Martín, M.B. Martínez Cruz, M. Martínez, B. Martínez, P. Martínez del Prado, R. Martino-Alba, M. Mayo, V. Mayoral, A. Meléndez, R. Mondéjar, M.A. Olalla Gallo, E. Ortega, M.L. Padilla del Rey, A. Pardo, J. Pérez Altozano, B. Pérez Benito, D. Pérez Murillo, P. Pérez Yuste, M.J. Redondo, J. Rocafort, D. Rodríguez, J.R. Rodríguez Mowbray, E. Rojo, J. Román, D. Ruiz-López, F.J. Sánchez Montero, N. Sánchez Martínez, S. Santaeugènia, V. Serrano, J. Trelis, A. Tuca, M. Vieito, J.A. Virizueta, and J. Zájara) for their participation.

**Conflicts of Interest:** C.P., T.Q., M.A.C., M.J.G., M.M., and A.I.G. acknowledge receiving grants from Medtronic Ibérica S.A. to conduct this project. T.Q. has received honoraria from Lilly, AstraZeneca, Daichi, Novartis, Roche, Rovi, and MSD outside the submitted work. I.P. has an educational contract with Medtronic Ibérica and Medtronic Europe and declares Medtronic stock ownership. M.A. is employed full-time by Medtronic Ibérica; S.A., M.C., and L.L. are employed full-time by Outcomes'10, a consultancy firm, which has received economic funding from Medtronic Ibérica, S.A. to develop this work. The authors hereby declare that this economic support has not interfered with the development of this project. The authors state that the sponsor did not participate or influence the analysis of the present study or the interpretations of its results. The authors have no other relevant affiliations or financial involvement with any organization or entity with a financial interest in or financial conflict with the subject matter or materials discussed in the manuscript apart from those disclosed.

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
