# Peer review of "Cancer-Related Pain Management in Suitable Intrathecal Therapy Candidates: A Spanish Multidisciplinary Expert Consensus"

_curroncol, doi:10.3390/curroncol30080530_

Round 1

Author Response

This paper presents a Delphi process related to intrathecal pain management in cancer patients. The study is well performed, and it is of interest that they have also included present status and expected difficulties with implementation in the process. The sample of respondents is large and representative to establish the opinions in Spanish health care. The discussion is adequate for the findings.

Thank you very much for your comment.

I have only two points that I believe needs to be considered.: The introduction do not present the potential down-sides with intrathecal therapy such as need to be attached to a syringe pump, need for close follow-up, risk for meningitis, risk for technical complications leading to loss of function, and risk for muscle paralysis. As it is now the authors give a biased presentation with only describing the benefits from intrathecal therapy.

According to reviewer suggestions, the following information regarding to intrathecal therapy has been included in the introduction:

“However, intrathecal therapy is not without risk. These complications can be procedure-related, device-specific or associated with the administered drug, and can range from catheter-tip granuloma to infections such as meningitis [11]”

I find the result section difficult to read and miss the actual statements. Without the statements I do not know to what the respondents agree or disagree. And usually, no reader bothers to look into supplementary. The table S2 is very good, and I will propose to replace all the results in part 3.2 including all figures with table S2. On a minor note I wonder if some of the statements in S2 could need some refinement in the translation from Spanish to English

Following reviewer comments, figures along the manuscript (Figure 1 to Figure 4) have been replaced by Table S2 (now Table 2).

We also confirm that the manuscript has been reviewed by a native English speaker.

Reviewer 2 Report

The use of interventional techniques is upcoming in cancer pain management. The use of intrathecal catheters is expecting to grow with increasing longlivety with metastatic disease. So the subject is certainly of interest.

However I have some serious questions with this paper: it doesn’t bring any new information. It mingles the necessity of multimodal pain treatment and open-door questions consensus about the importance of working together, having enough experience and also taking psychosocial factors into account.

And having mentioned the importance of a team: GP’s and community nurses are missing in all Delhi rounds and they are extremely important in making it a success. You mention them in the only discussion as very important: “especially when considering both hospital and home care”.

There are more review articles about intrathecal drug delivery than actual studies. The references used in this paper are an example of that.

Reference 1-3 are from reviews: use the original studies, reference 5 should be the WHO effectiveness and not another ITDD paper, reference 6: Mercadante says: 10 – 20 % and not approximately 20% (exaggerating). Reference 8 is a review of only non-cancer pain. Also in the discussion reference8-11 are reviews, only ref 18 is a (retrospective) study.

You suggest a controversy in paragraph 4 an 5 of the introduction: I don’t see it.

The context of the discussion is very Spanish, that’s oké, but please mention it in the title.

Author Response

The use of interventional techniques is upcoming in cancer pain management. The use of intrathecal catheters is expecting to grow with increasing longlivety with metastatic disease. So the subject is certainly of interest.

However I have some serious questions with this paper: it doesn’t bring any new information. It mingles the necessity of multimodal pain treatment and open-door questions consensus about the importance of working together, having enough experience and also taking psychosocial factors into account.

And having mentioned the importance of a team: GP’s and community nurses are missing in all Delhi rounds and they are extremely important in making it a success. You mention them in the only discussion as very important: “especially when considering both hospital and home care”.

There are more review articles about intrathecal drug delivery than actual studies. The references used in this paper are an example of that.

Thank you for your comments. We aim to provide recommendations, agreed upon by a group of experts, about managing oncological pain in those patients who are potential candidates for intrathecal therapy. These recommendations may guide healthcare professionals to optimize the management of these patients. 

Regarding study participants, we confirm that both community nurses and GPs have participated as members of the Delphi panel. In order to clarify this aspect, the sentence has been rewritten.

“Healthcare professionals with extensive experience in cancer pain management were selected as panelists and invited to participate. Pain unit anesthesiologists, oncologists, palliative care specialists, nurses involved in cancer pain treatment (including hospital and community nurses), home hospitalization unit specialists (including primary care physician), and case managers, were included.”

Reference 1-3 are from reviews: use the original studies, reference 5 should be the WHO effectiveness and not another ITDD paper, reference 6: Mercadante says: 10 – 20 % and not approximately 20% (exaggerating). Reference 8 is a review of only non-cancer pain. Also in the discussion reference8-11 are reviews, only ref 18 is a (retrospective) study.

According to your comments, the following references have been changed:

  • References 1-3 have been replaced by original studies:
    • Snijders RAH, Brom L, Theunissen M, van den Beuken-van Everdingen MHJ. Update on Prevalence of Pain in Patients with Cancer 2022: A Systematic Literature Review and Meta-Analysis. Cancers (Basel). 2023;15(3).
    • Carr D, Goudas L, Lawrence D, Pirl W, Lau J, DeVine D, et al. Management of cancer symptoms: pain, depression, and fatigue. Evid Rep Technol Assess (Summ). 2002(61):1-5.
  • Reference 5 has been changed by WHO report:
    • WHO Guidelines for the Pharmacological and Radiotherapeutic Management of Cancer Pain in Adults and Adolescents. Geneva2018.
  • Reference 6: has been changed in the text:
    • “However, it is estimated that from 10 to 20% of patients are unresponsive or unable to tolerate the conventional first-line approaches and, therefore, require interventional treatment [6]”
  • Reference 8: this reference is a review of chronic pain management with intrathecal therapy. Although it is not specific to oncological patients, it does not exclude them either. We consider this reference appropriate for the article, given that the aspect it deals with also applies to oncological patients.
  • Reference 8 to 11: we agree that these references are reviews, but they are reviews of the currently available evidence, so we consider them appropriate to include in the discussion.

You suggest a controversy in paragraph 4 an 5 of the introduction: I don’t see it.

The context of the discussion is very Spanish, that’s oké, but please mention it in the title.

According to the reviewer suggestion, Tittle has been modified to mention the context of the study:  “Cancer-related pain management in suitable intrathecal therapy candidates: a Spanish multidisciplinary expert consensu”
